# Pubo-Femoral Distances Measured Reliably by Midwives in Hip Dysplasia Ultrasound

**DOI:** 10.3390/children9091345

**Published:** 2022-09-02

**Authors:** Hans-Christen Husum, Michel Bach Hellfritzsch, Rikke Damkjær Maimburg, Mads Henriksen, Natallia Lapitskaya, Bjarne Møller-Madsen, Ole Rahbek

**Affiliations:** 1Interdisciplinary Orthopaedics, Aalborg University Hospital, 9000 Aalborg, Denmark; 2Danish Paedatric Orthopaedic Research, Aarhus University Hospital, 8200 Aarhus, Denmark; 3Department of Radiology, Aarhus University Hospital, 8200 Aarhus, Denmark; 4Department of Obstetrics and Gynecology, Aarhus University Hospital, 8200 Aarhus, Denmark; 5Department of Children’s Orthopaedics, Aarhus University Hospital, 8200 Aarhus, Denmark

**Keywords:** developmental dysplasia of the hip, ultrasound, mass screening

## Abstract

The pubo-femoral distance (PFD) has been suggested as an ultrasound screening tool for developmental dysplasia of the hip (DDH). The aim of this study was to examine if midwives undergoing minimal training could reliably perform pediatric hip ultrasound and PFD measurements. Eight recruited midwives performed two rounds of independent blinded PFD measurements on 15 static ultrasound images and participated in four supervised live-scanning sessions. The midwives were compared to a group of three experienced musculoskeletal radiologists. Reliability was evaluated using inter-rater correlation coefficients (ICC). Linear regression was used to quantify the learning curve of the midwives as a group. There was near complete intra- and inter-rater agreement (ICC > 0.89) on static ultrasound images across both rounds of rating for midwives and radiologists. The midwives performed a mean of 29 live hip scans (range 24–35). The mean difference between midwives and supervising radiologists was 0.36 mm, 95% CI (0.12–0.61) for the first session, which decreased to 0.20 mm, 95% CI (0.04–0.37) in the fourth session. ICC for PFD measurements increased from 0.59 mm, 95% CI (0.37–0.75) to 0.78 mm, 95% CI (0.66–0.86) with progression in sessions. We conclude that midwives reliably perform PFD measurements of pediatric hips with minimal training.

## 1. Introduction

Developmental dysplasia of the hip (DDH) is a condition of underdevelopment of the hip joint and ranges from mild acetabular shallowing to complete hip dislocation. With an incidence of 0.8%, it is the most common musculoskeletal disorder in children [1]. Treatment of DDH is time-sensitive as early diagnosed cases can be treated conservatively and successfully with hip bracing [2], and delayed diagnosis worsens the prognosis and increases the necessity of surgery and risk of complications [3]. Because of the relatively high incidence, and the time-sensitive nature of the condition, ultrasound screening has been widely implemented in high-resource countries. The predominant screening strategy is selective ultrasound screening based on the Graf ultrasound method [4]. At-risk newborns receive a hip ultrasound based on clinical examinations of the hip or the presence of risk factors for DDH. In contrast, in universal ultrasound screening every child receives a hip ultrasound regardless of clinical findings and/or risk factors. The only Cochrane review on DDH screening to date was inconclusive in whether to recommend selective- or universal ultrasound screening [5] due to a lack of decisive evidence. Since the Cochrane review was published, evidence increasingly points towards limitations in the selective screening approach, as 85% of patients treated for DDH do not fulfill the criteria for selective ultrasound screening [6]. Further, primary screeners have insufficient knowledge of the clinical examinations and risk factors which constitute the foundation of the selective screening approach [7,8,9]. These findings might explain why 26 years of selective ultrasound screening in the UK has failed to reduce the rate of late DDH diagnoses. [10]. Since the 1990s, universal DDH ultrasound screening of all newborns with the Graf method has been performed in parts of Austria and Germany, resulting in near-eradication of late DDH diagnoses [1], and in the lowering of treatment costs although diagnostic expenses have increased similarly [11]. The international consensus on DDH screening is now shifting towards universal DDH ultrasound screening based on Graf’s method [12].

A central obstacle in the implementation of universal Graf ultrasound screening is, however, the increase in diagnostic expenses and demand for specialized radiological resources in part due to the complexity of the Graf ultrasound method. In 2013 the pubo-femoral distance (PFD) was proposed as an alternative, less complex ultrasound measurement for DDH screening [13] with a high sensitivity and specificity for DDH [14]. The PFD is defined as the minimal, measurable distance between the medial femoral epiphysis and the pubic bone while applying lateralizing stress to the hip. Minimal experience is necessary to perform the measurement reliably [13]. However, the approach has only been documented when performed by radiologists. To reduce the need for specialized radiological resources and in turn the economic impact of implementing a universal PFD screening program, we hypothesized that the PFD method can be taught with minimal instruction to non-physician health-care professionals with little or no prior experience in ultrasonography while still observing a high degree of accuracy. For this purpose, midwives were selected as they are the health care professionals making the primary clinical examination of the newborn in the Danish neonatal screening program. 

Our aim was firstly to demonstrate the agreement in midwife PFD measurements on static and live ultrasound images compared with measurements performed by experienced musculoskeletal radiologists. Secondly, our aim was to quantify the learning curve for the PFD ultrasound measurement among midwives.

## 2. Materials and Methods

### 2.1. Study Design

This was a prospective observational study. Reporting follows the STROBE guidelines for reporting on observational studies [15].

### 2.2. Participants

We recruited midwives for training in PFD measurement from the Department of Obstetrics and Gynecology at Aarhus University Hospital, Denmark. Recruited midwives completed a demographic survey which included experience as a midwife, prior experience in ultrasound examinations, and number of yearly clinical DDH screenings performed. For comparison, three musculoskeletal radiologists were recruited from the Department of Radiology at Aarhus University Hospital, with respectively, 21, 7 and 1.5 years of experience in pediatric hip ultrasound examinations. 

### 2.3. Constructing the PFD Training Program

The midwives’ training program was conceptualized on the theoretical background described in Miller’s pyramid [16] to achieve professional clinical competence. The learning content was developed on the principles of blended learning to provide both different learning styles and learning environments. We started the training program with an online introduction film about hip dysplasia followed by a traditional theoretical lecture advancing with subsequent on-site practical demonstration and exercises with online introduction and instructional videos. 

### 2.4. Lectures and Workshops

Each midwife was instructed to watch a 10-min online video introduction on DDH and PFD screening prior to participating in a two-hour theoretical group lecture on basic anatomy, pathogenesis and treatment of DDH as well as an introduction to pediatric hip ultrasound, with an emphasis on the PFD measurements and video demonstrations of how to obtain it. Afterwards, each midwife participated in two workshops. In the first workshop, the midwives were evaluated in a best-case scenario, where they were presented with best-practice static hip ultrasound images in the standard projection according to Graf without annotations, obtained by a senior musculoskeletal radiologist with 21 years of experience. Each midwife performed PFD measurements on 15 images using Picture Archiving and Communication Software (PACS) (Impax, client 6.5 AGFA Healthcare, Mortsel, Belgium). Seven days later, in the second workshop, the participants repeated these measurements on the same images. For comparison, the recruited radiologists performed the same measurements with seven-day intervals. The measurement exercises of the midwives were monitored by the first author, and no instructions other than technical support were provided. All raters were blinded to the measurements performed previously by themselves and others and they were instructed not to share information on measurements. 

### 2.5. Supervised Live-Scanning Sessions

After completing the lecture and workshops, each midwife received a 30-min introduction to the MINDRAY TE7 (Mindray Medical International, Shenzhen, China) ultrasound scanner as well as a brief introduction to general sonography. Each midwife then participated in four sessions of supervised live scans of pediatric hips over the course of two weeks as part of the DDH screening program at our institution where PFD measurements are routinely measured. Scanning sessions were supervised by one of the three senior musculoskeletal radiologists and integrated into the current ultrasound screening program for DDH in the Radiological Outpatient Clinic at our institution. The live-scanning sessions took place between September 2021 and December 2021. For the purpose of this study, a separate MINDRAY TE7 ultrasound scanner with a simplified user-interface and high frequency (16 MHz) linear transducer was acquired and calibrated specifically for pediatric hip ultrasound. In accordance with the institution protocol, the radiologist performed a hip ultrasound on one side with the newborn in the lateral decubitus position according to the method described by Graf, Tréguier and Couture [4,13,17] using a 10 MHz linear transducer (Model: Canon Aplio i800; Canon Medical Systems, Tokyo, Japan). The midwife then repeated the scan using the MINDRAY TE7 ultrasound scanner and measured the PFD. Finally, to avoid bias introduced by using two different scanners, the radiologist would perform the PFD measurement using the MINDRAY TE7 scanner. The entire sequence was then repeated for the opposite hip. 

The criteria for the ultrasound scan were initially visualization of the femoral head and the lateral epiphysis of the pubic bone while adducting the knee and performing a Barlow equivalent lateralizing stress maneuver on the hip joint to perform the PFD measurement. As the midwife gained experience, to ensure consistency and repeatability, the criteria were expanded to include the horizontal plane of ilium and the bony and cartilaginous acetabular roof in accordance with the method described by Tréguier and Couture [13,17] (Figure 1).

As these sessions also functioned as a training regimen in pediatric hip ultrasound for the midwives, the radiologists were free to instruct the midwives and give feedback as needed until the midwife had performed the PFD measurement. Neither the midwife nor the radiologist was blinded to the measurement results but they did not make repeat PFD measurements of the same hip and were instructed to disregard the measurements of the other party. The first author was present to enforce these instructions.

### 2.6. Data Sources

Static hip ultrasound images without annotations for the PFD measurement workshop were acquired from the production of the existing ultrasound screening program at our institution, and all were performed by a senior musculoskeletal radiologist according to the methods of Graf, Tréguier and Couture [4,13,17]. All measurements from the workshops and live-scanning sessions were anonymized and registered directly by the first author in a General Data Protection Regulation compliant REDCap database.

### 2.7. Statistical Methods

#### 2.7.1. Workshop Measurements

We analyzed intra- and inter-rater reliability of PFD measurements within and between the group of radiologists and midwives using intraclass correlation coefficients (ICC) with accompanying 95% confidence intervals (CI). Intra-rater ICCs were calculated as two-way mixed effects, single measurement with absolute agreement, while two-way random effect was used for the inter-rater ICCs. We interpreted ICCs according to Portney and Watkins [18] with a value of 0, 0.75 and 1 representing no agreement, good agreement and complete agreement, respectively.

#### 2.7.2. Supervised Live-Scanning Sessions

To investigate any correlation in agreement with increasing PFD values, and to illustrate the overall progress in agreement between midwives and radiologists with increasing measurement experience among the midwives, a Bland–Altman (BA) plot for each session was made with mean difference and limits of agreement (LOA). A scatter plot was made of the absolute differences in PFD values between radiologists and midwives as a function of improved experience among the midwives, defined as the cumulative number of patients scanned. The scatter plot included the mean absolute differences for the midwives as a group, as well as a fitted linear regression with 95% CI. To quantify the agreement as the midwives gained experience, a linear regression was performed with absolute difference as the dependent variable and number of patients scanned per midwife as the explanatory variable while controlling for previous ultrasound experience as a dichotomous yes/no variable. Normal distribution of PFD differences was inspected using QQ plots and a significance level of 5% was applied. Statistical analyses were performed using Stata version 16.1 (StataCorp, College Station, TX, USA).

### 2.8. Ethics

This was a quality control study which followed the routine DDH screening program at our institution. Findings had no impact on patient treatment or diagnosis. As per the guidelines from the Danish National Center for Ethics, ethical approval and written consent was not needed. 

## 3. Results

Eight midwives were included in this study. Mean years of seniority as a midwife was 11 years (range 4–27 years), mean number of yearly clinical DDH screening was 68 (range 10–110). Two midwives performed fetal ultrasound in clinical practice and had done so for one year (Table 1).

### 3.1. Workshops

There was near complete intra- and inter-rater agreement (ICC > 0.89) in PFD measurements of static ultrasound images in both workshops across and within the midwife and radiologist groups. No difference was found in ICC between the groups of midwives and radiologists or across rounds of rating (Table 2).

### 3.2. Supervised Live-Scanning Sessions

The mean number of hips scanned by the midwives over the four supervised live-scanning sessions was 29 (range 24–35) with a total of 237 hips scanned. Inspection of BA plots did not reveal any dependency between differences in PFD and mean PFD values. In the first supervised session the mean difference was 0.36 mm (LOA −1.42; 2.14 mm) which decreased to 0.2 mm (LOA −1.04; 1.44 mm) in the fourth and final supervised session (Figure 2).

Inter-rater agreement between midwives and radiologists increased from ICC = 0.59 with a 95% CI (0.37; 0.75) in the first supervised session to ICC = 0.78 (0.66; 0.86) in the third supervised session. ICC for the fourth supervised session could not be evaluated due to low variance in observed values (Table 3).

Scatter plot inspection revealed a decrease in the range of absolute PFD differences between midwives and radiologists as the midwives became more experienced (Figure 3). Initial mean absolute PFD difference between midwives and radiologists was 0.73 mm, which decreased by 0.1 mm (95% CI 0.02–0.17 mm) for every ten scans the midwives gained in experience (*p* < 0.008) and was not associated with previous ultrasound experience of the midwife (*p* = 0.51).

## 4. Discussion

In this study, midwives’ performance in PFD measurements on static ultrasound images showed the same level of reliability as measurements performed by senior musculoskeletal radiologists, following minimal instruction of midwives in PFD measurement by ultrasound. After a short learning program, including three supervised sessions, midwives reliably performed pediatric hip ultrasound and PFD measurements with clinically insignificant differences when compared to experienced radiologists. 

### 4.1. Limitations

The results may be affected by observer- and performance bias, as both groups were aware that their measurements would be logged, and the midwives, although instructed to disregard the radiologists’ measurements, were not blinded to these while doing live scans. These biases may result in an observed higher agreement in observations. Due to low variance in observed values in session four, ICC values could not be calculated, which is a known statistical limitation of the ICC method [19]. However, the Bland–Altman plot depicts larger agreement in PFD measurements between groups with increasing session numbers including session four, which is also reflected in the calculated decrease in absolute PFD difference between groups as the midwives became more experienced.

### 4.2. Interpretation

It is of vital importance for imaging-based screening protocols to be reliable and accurate. To date, two studies have examined the reliability of the PFD method. Teixeira et al. compared PFD measurements of one senior radiologist to one resident radiologist and found a high degree of inter-rater agreement in measurement values (ICC = 0.85) [20]. Tréguier et al. compared one senior radiologist to a radiologist in training and documented a mean measurement difference of 0.12 mm and found a substantial agreement in categorizing hips with a threshold of 6 mm (Kappa = 0.795) [13]. The present study, is the first to compare examiners with no previous experience in hip ultrasound to a group of experienced musculoskeletal radiologists, and we found similarly high levels of agreement. It is worth noting that mean measurement difference is a poor estimate for agreement, as a low value could mean that the measurements are evenly under- and overestimated. In contrast, absolute differences give a more precise estimate of agreement as they are not affected by over- and underestimations and should be used when reporting on agreement in measurements.

In a recent meta-analysis of ultrasound measurements used in DDH diagnostics, considerable variation was found for Graf’s alfa and beta angle, as well as Terjesen’s Femoral Head Coverage (FHC). Reliability was poor to moderate for all ultrasound-based metrics, and although variation was lowest for the alfa and beta angles, intra- and inter-rater agreement still varied from an ICC of 0.02 to 0.453. Further, reliability of the Graf classification of hips in the included studies was poor to moderate (Kappa 0.1–0.6) [21]. The relatively high complexity of the Graf measurements and the susceptibility to interpretation errors due to mispositioning of the ultrasound probe may lead to misinterpretation of measurements and consequently misclassification of healthy hips as dysplastic and vice versa. Jaremko et al. were able to produce clinically acceptable hip ultrasound images with a 20 degree tilt error in probe positioning, causing an alpha angle variation of 18 degrees (52–70 degrees) leading to misclassification in 54% of the hips scanned [22]. Orak et al. investigated the reliability of the alfa and beta angles and found poor inter-rater agreement among four raters with experience from >500 hip ultrasound examinations [23]. Despite the extensive experience of the raters and a pre-study consensus meeting, images from this study depicting four different interpretations of the same hip, showed inconsistencies in the application of the Graf method, translating to alpha angles ranging from 57–72 degrees or normal to mild hip dysplasia. In contrast, the PFD method is a simple single distance measurement between two distinct landmarks. Although the influence of probe positioning on PFD measurements has not yet been established, the reported high levels of inter-rater agreement in this study, among users with varying hip ultrasound experience, suggest a higher tolerance for tilt and rotational errors of the probe.

A key issue in implementing universal ultrasound screening for DDH is the increase in diagnostic expenses and an increased demand for experienced ultrasound examiners. This is largely due to the experience needed to perform and interpret pediatric hip ultrasound using the Graf method [11]. The PFD method has been suggested as an alternative to the Graf method as a highly sensitive measurement for DDH [13,14]. As demonstrated by the present study, PFD screening measurements performed by novices did not result in lowering accuracy, thus demonstrating a possible cost-effective alternative to current screening practices, and supporting the feasibility of employing midwives to perform PFD ultrasound screening for DDH. Further studies on PFD ultrasound screening programs using non-radiologist examiners are, however, needed to evaluate the efficacy in detecting DDH through PFD screening, based on trained novice ultrasound users. 

#### Generalizability

The degree of dysplasia based on Graf’s method was not registered for the infants scanned in this study. However, as we consider a PFD > 6.0 mm to be indicative of DDH [13], and we did not find an increase in PFD between raters with increasing mean PFD values, we believe these results are valid in a dysplastic population. The midwives in this study were diverse in terms of years of seniority, number of yearly clinical DDH screenings and previous ultrasound experience, with the majority having no experience in performing ultrasound examinations. We therefore expect these results to be applicable to health care professionals with no experience in ultrasound.

## 5. Conclusions

We conclude that midwives, undergoing theoretical and limited supervised practical training, are able to perform PFD measurements of pediatric hips with the same level of reliability and precision as senior musculoskeletal radiologists.

## Figures and Tables

**Figure 1 children-09-01345-f001:**
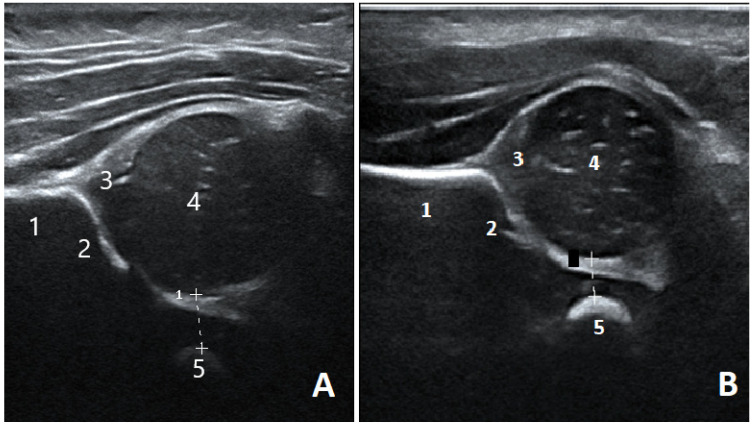
Ultrasound image of a two different newborn hips obtained by a musculoskeletal radiologist (**A**) and a midwife (**B**) depicting the quality criteria for the PFD measurement: A horizontal ilium (1), the bony (2) and cartilaginous (3) acetabular roof, the femoral head (4) and the lateral epiphysis of the pubic bone (5). The PFD is the minimal distance between the medial femoral epiphysis and the pubic bone (dotted line). PFD = Pubo-femoral distance.

**Figure 2 children-09-01345-f002:**
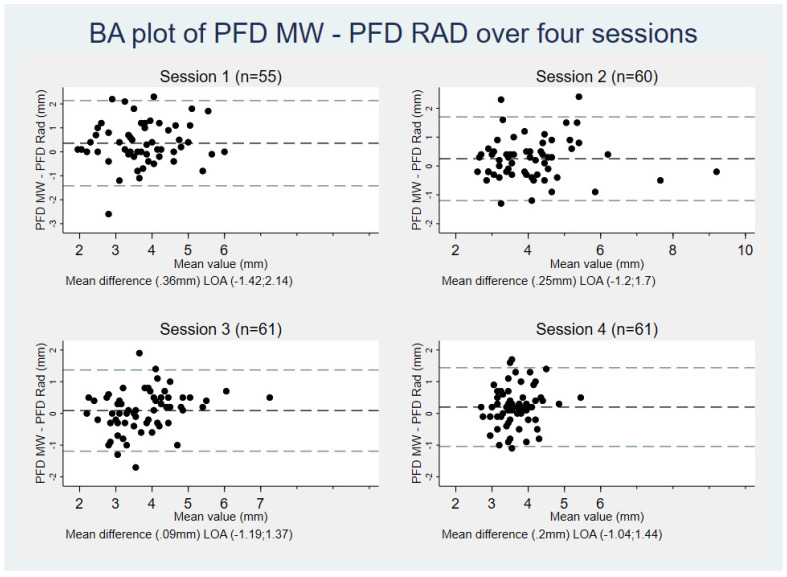
Bland–Altman plot of differences in PFD measurements between radiologists and midwives across four sessions of supervised pediatric hip scans. PFD = Pubo-femoral distance, MW = Midwife, RAD = Radiologist, LOA = Limits of agreement.

**Figure 3 children-09-01345-f003:**
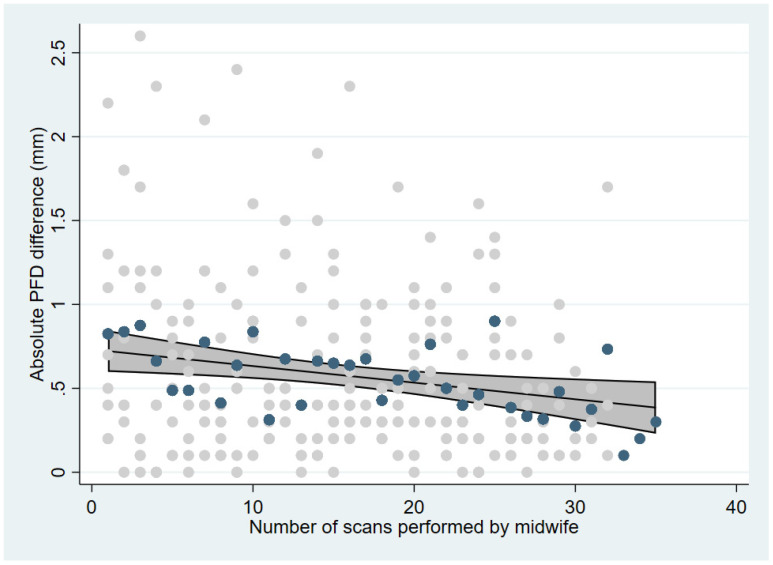
Scatter plot and fitted linear regression with 95% CI of absolute differences in PFD measurements between radiologists and midwives as a function of increasing scan experience. Light grey dots = individual values, blue dots = average values. PFD = Pubo-femoral distance.

**Table 1 children-09-01345-t001:** Demographics of recruited midwives.

Years of seniority as a midwife (mean (range))	11 years (4 years–27 years)
Number of clinical DDH screenings (Ortolani and Barlow maneuvers) yearly (mean (range))	68 (10–110)
Uses ultrasound in clinical practice (fetal ultrasound) (yes/no)	2/6
Years of experience using ultrasound in clinical practice (*n* = 2) (range)	1 (1)

**Table 2 children-09-01345-t002:** Inter-rater and intra-rater ICCs of PFD measurements made on 15 static pediatric hip ultrasound images across two rating workshops. ICC = Inter-rater correlation coefficient, PFD = Pubo-femoral distance, RAD = Musculoskeletal radiologist, MW = Midwife.

	Radiologists (*n* = 3)	Midwives (*n* = 9)
Inter-rater RAD/MW ICC (95% CI) workshop 1	0.99 [0.86;0.99] 0.99 [0.92;0.99]	
Inter-rater RAD/MW ICC (95% CI) workshop 2		
Inter-rater ICC (95% CI) within group workshop 1	0.93 [0.84;0.97]	0.89 [0.80;0.95]
Inter-rater ICC (95% CI) within group workshop 2	0.95 [0.83;0.98]	0.95 [0.90;0.98]
Intra-rater ICC (95% CI) group average between workshops	0.98 [0.93;0.99]	0.99 [0.84;0.99]

**Table 3 children-09-01345-t003:** Inter-rater ICCs of PFD measurements made by recruited midwives and supervising radiologists across four sessions of live scans. ICC = Inter-rater correlation coefficient.

Session of Live Scans (*n* = Hips)	ICC (95% CI)
Session 1 (*n* = 55)	0.59 (0.37;0.75)
Session 2 (*n* = 60)	0.81 (0.68;0.88)
Session 3 (*n* = 61)	0.78 (0.66;0.86)
Session 4 (*n* = 61)	0.42 (0.19;0.60) *

* Due to an insufficient variation in measured values, the calculated ICC value falsely quantified reliability in session 4 as low. This does not reflect the high level of agreement between the two groups of raters as can be seen from the BA plot and limits of agreement (Figure 2). The underestimation of ICC values due to low variation of observations is a known limitation of the ICC method [19], the value is only presented here for transparency.

## Data Availability

A notification on the data set and the possibility of gaining access will be shared to Aalborg University’s research portal (VBN.aau.dk).

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
