# Peer review of "Pubo-Femoral Distances Measured Reliably by Midwives in Hip Dysplasia Ultrasound"

_children, 2022, doi:10.3390/children9091345_

Round 1
Reviewer 1 Report
First of all, I would like to congratulate the authors on their research.
My comments:
As a whole, the article needs reworking and rephrasing. Different structure as this version with many sub-titles and new lines in the text is rather distracting. Also, there should not be two discussion sections.
Introduction:
A comment on difficulties in clinical vs radiological examination and their impact on patients’ future should be mentioned in the introduction
Lines 55-56 need rephrasing ... cost-costs
Materials methods:
Do you think online lectures and lather two workshops were enough education for the base knowledge of sonography?
Lines 154-155 as midwives were instructed to disregard measurements by other parties, which could produce a significant bias as no trainees do tend to change results upon finding any unexpected errors in their work, this is different for professionals and experienced personnel.
Did they actually have access to these measurements or were they present when these took place?
Results:
Midwives with prior usg experience – specify the field and level of experience.
-did these midwives show better results in hip usg?
Did the group of radiologists have deviations in their measurements and did they have deviations in measurements on the two different sonographs used?
A decrease in the difference between RAD and MW groups in the four sessions is visible, but the fourth session difference rose again. What could be the cause?
The question is if RAD measurements were consistent in all sessions and what kind of a difference did these three radiologists produce on the same hips.
Discussion:
- There are two discussion sections, this needs correction
The mean year of seniority does not have to be mentioned in results and also in discussion, read through the text and remove obsolete duplications. Also, this information could rather be included in materials and methods, but I leave this to the authors' decision.
The interpretation part could be part of a significantly reworked discussion.
Author Response
First of all, I would like to congratulate the authors on their research.
My comments:
As a whole, the article needs reworking and rephrasing. Different structure as this version with many sub-titles and new lines in the text is rather distracting. Also, there should not be two discussion sections.
- Thank you for a valid point. We prepared the manuscript per the journal template. The different sub-titles are used to highlight the two different data sources (measurements on static images in the training workshop and measurements on live supervised scans.)
- Per your suggestions we have significantly reworked the structure of the manuscript to avoid line changes and single sentence paragraphs.
- The duplicate discussion is a regrettable error and has now been removed.
Introduction:
A comment on difficulties in clinical vs radiological examination and their impact on patients’ future should be mentioned in the introduction
- We agree that this topic is extremely relevant, but chose to limit the scope of the introduction to radiological screening, specifically universal ultrasound and why we have chosen to pursue the PFD method. The suggested discussion by the reviewer could warrants its own research article, and we would certainly encourage that.
Lines 55-56 need rephrasing ... cost-costs
- Has now been rephrased
Materials methods:
Do you think online lectures and lather two workshops were enough education for the base knowledge of sonography?
- We failed to mention that each participating midwife also received a 30min introduction to the MINDRAY TE7 ultrasound scanner and a general introduction to sonography. This has now been added to the manuscript under methods (line 115).
- Regardless we agree with the reviewer that this is not enough for the base knowledge of sonography, but this was not our intention.
- We took a pragmatic approach in omitting teaching of sonography concepts and focusing entirely on the application of the PFD method with the sole purpose of developing a rapid teaching program to be used in DDH screening. We believe the present article demonstrates that our teaching program was sufficient to enable novice ultrasound users to only measure PFD in pediatric hips, but we do not suggest that we have trained the participants to perform any other form of sonography.
Lines 154-155 as midwives were instructed to disregard measurements by other parties, which could produce a significant bias as no trainees do tend to change results upon finding any unexpected errors in their work, this is different for professionals and experienced personnel.
Did they actually have access to these measurements or were they present when these took place?
- The midwives were present while the radiologist performed the initial ultrasound on one side and were instructed to disregard the measurements obtained. They would then repeat the measurement on the same side and the same process was then repeated for the opposite hip (line 126-133).
- This could bias the results towards a higher agreement (as mentioned in the discussion), but was unavoidable as this was also the training program for the midwives. To limit the impact, the first author was present at all supervised sessions to instruct the midwives to ignore the initial measurements performed by the radiologist. (line 141-146)
Results:
Midwives with prior usg experience – specify the field and level of experience.
- Table 1 has been rephrased to specify the field as fetal ultrasound
- Fetal ultrasound has been added (line 193)
- Level of experience depicted in table 1 as years of experience with range and in results (line 193-194)
-did these midwives show better results in hip usg?
- There was no statistical correlation of observed agreement to previous usg experience (line 231-232)
Did the group of radiologists have deviations in their measurements and did they have deviations in measurements on the two different sonographs used?
- Thank you for this valuable question. The group of radiologists were only directly compared using the measurements obtained from the static ultrasound images, as only one of the three radiologist were present at the supervised sessions. ICC values for agreement between radiologists are presented in table 2. There was near complete agreement for all measurements (ICC >0.90).
- The present study did not compare the measurements between ultrasound machines. To eliminate any measurement error introduced by the sonograph used, all live measurements presented in this article are performed using the MINDRAY TE7 ultrasound scanner (line 131-133).
A decrease in the difference between RAD and MW groups in the four sessions is visible, but the fourth session difference rose again. What could be the cause?
- A good observation. The mean difference rose in session 4 but the limits of agreement narrowed. The difference in mean difference between session 3 and 4 was not statistically significant.
- This highlights the limitation of mean difference as an estimate when comparing agreement, as low mean difference simply indicates that the measurements are evenly distributed both ways, but provides no information on the degree of error. We chose only to report mean difference accompanied with BA plot to illustrate how the distribution narrowed with increasing sessions.
- In contrast, absolute differences are depicted in figure 3 which gives a more precise impression of the development in agreement over time.
- We have added a paragraph discussing the limitations of the mean measurement for agreement to the discussion section (line268-272).
The question is if RAD measurements were consistent in all sessions and what kind of a difference did these three radiologists produce on the same hips.
- A very good question which has prompted us to revise the manuscript as it was not made clear that the PFD measurements are routinely obtained at our institution by the radiologists and as such have extensive experience with the measurement. This has now been added to the methods section (line 119 + line 128)
- A comparison of agreement in PFD measurements between radiologists have previously been made and shown to be highly consistent (Line 62 + reference 13). A comparison of RAD live measurements was therefore not recorded nor included in this study.
Discussion:
- There are two discussion sections, this needs correction
- A regrettable formatting error. The duplicate discussion section has now been removed.
The mean year of seniority does not have to be mentioned in results and also in discussion, read through the text and remove obsolete duplications. Also, this information could rather be included in materials and methods, but I leave this to the authors' decision.
- Years of seniority are presented in results per the STROBE reporting guidelines as demographic results of the participants. We do not consider the mentioning of seniority in the discussion as obsolete as it is a central part of the external validity/generalizability of the study, and as such is featured in the “Generalizability” section, which is also dictated by the STROBE guidelines.
- We chose to include this data to illustrate the variety of midwives involved in terms of seniority as to discourage suspicions of cherry-picking experienced participants for teaching.
The interpretation part could be part of a significantly reworked discussion.
- Per the reviewer’s excellent suggestions, the discussion section has been reworked (see above).
Reviewer 2 Report
However, the method that has been described is very uncommon, yet as it was described in the introduction, it`s costs are favorable to Graf`s method.
In my opinion the group is much to low for such a research. The originality is average - it is all about learning curve.
Author Response
However, the method that has been described is very uncommon, yet as it was described in the introduction, it`s costs are favorable to Graf`s method.
- We agree. The PFD method is relatively new (2013) and as such is not widely implemented in DDH screening outside the original authors’ region of France.
- We routinely measure PFD as part of the DDH screening program at our institution. However, PFD values do not yet guide treatment- or diagnostic decision making.
In my opinion the group is much to low for such a research. The originality is average - it is all about learning curve.
- Thank you for this comment. The secondary aim of the article was indeed to show the learning curve. It is our opinion that the primary aim (agreement) is well reflected in the outcomes.
- With regards to the originality we disagree. Documentation of a swift learning curve is of vital importance if a universal ultrasound screening program is to be pursued as it indicates the cost-effectiveness of such a program and is a central argument for any mass screening program.
Reviewer 3 Report
Title: Pubo-femoral distances measured reliably by midwives in hip dysplasia ultrasound.
Thank you for the opportunity to review this paper. It was very interesting and has the potential to add to the diagnosis and care of DDH. Good work.
Introduction
Overall excellent content for an introduction explaining the background the holes in the literature and clinical setting and why the aim/hypothesis is justified. It may just be a bit long and requires some grammatical and structural changes.
- overall mild grammatical issues throughout the paper i.e. line 32 '...of 0.8%, it is the...'
Additionally consider writing paragraphs rather than bullet points for the introduction. One sentence points look awkward and unprofessional.
- line 64 - reference 14 needs to be displayed
- personally I believe figures such as Figure 1 should come from the methods section as the measurement details are described then.
Methods
Very comprehensive and clear methods.
Was ethics really deemed unnecessary? I find this hard to believe given it was a prospective study with teaching/imaging involved in a new born by multiple people.
Similar to the introduction the bullet point style "statistics" section should be 1 paragraph for each subheading. i.e line 165 - 171 together.
Results
clearly explained and displayed
Discussion
Clear and too the point. Unsure about subheadings in discussions.
as you are likely aware it appears your results have doubled up from line 235 to 267. please remove
line 271 - PFD not PDF.
Once again it would be good to make paragraphs. i.e. group line 269-275. This principle should likely be implemented throughout the paper.
Author Response
Thank you for the opportunity to review this paper. It was very interesting and has the potential to add to the diagnosis and care of DDH. Good work.
Introduction
Overall excellent content for an introduction explaining the background the holes in the literature and clinical setting and why the aim/hypothesis is justified. It may just be a bit long and requires some grammatical and structural changes.
- Thank you for generous comment. We have rephrased parts of the introduction and improved grammar.
overall mild grammatical issues throughout the paper i.e. line 32 '...of 0.8%, it is the...'
- Manuscript has been reworked to remove grammatical errors where spotted.
Additionally consider writing paragraphs rather than bullet points for the introduction. One sentence points look awkward and unprofessional.
- Thank you for this suggestion. The manuscript has been restructured to avoid single sentence segments.
line 64 - reference 14 needs to be displayed
- Reference 14 is now properly displayed
personally I believe figures such as Figure 1 should come from the methods section as the measurement details are described then.
- Thank you for the suggestion. Figure 1 has been moved to the methods section.
Methods
Very comprehensive and clear methods.
Was ethics really deemed unnecessary? I find this hard to believe given it was a prospective study with teaching/imaging involved in a new born by multiple people.
- The study was done as part of the routine DDH screening program at our institution (now added as line 119), where PFD measurements are routinely measured although they do not guide diagnostic or treatment decisions. As such the PFD measurements had no impact on decision making and the study was registered as a quality control study.
- Per the guidelines from the Danish Committee for health research ethics. Ethics approval and written consent was therefore not necessary. A link to the guideline (unfortunately in Danish) has been added to the Ethics section of the paper, per the instructions from the editor.
- Ethics section has been edited to reflect the answer given above.
Similar to the introduction the bullet point style "statistics" section should be 1 paragraph for each subheading. i.e line 165 - 171 together.
- Thank you for this point. “Bullet point” style has been reworked in the entire manuscript.
Results
clearly explained and displayed
Discussion
Clear and too the point. Unsure about subheadings in discussions.
- Thank you once more for this generous comment. The subheading of the discussion reflect the STROBE reporting guidelines. If it is the reviewers’ suggestion that these should be removed, we will be happy to do so if the above explanation is unsatisfactory.
as you are likely aware it appears your results have doubled up from line 235 to 267. please remove
- A regrettable formatting error. The duplicate Discussion section has now been removed.
line 271 - PFD not PDF.
- Indeed! This has now been corrected.
Once again it would be good to make paragraphs. i.e. group line 269-275. This principle should likely be implemented throughout the paper.
- See above.
Round 2
Reviewer 2 Report
The revised version provides sufficient information and background for the method used in the article. A few sentences about Graf`s method ensured sufficient context.